# Better Language Model Inversion by Compactly Representing Next-Token Distributions

**Murtaza Nazir**[*]  **Matthew Finlayson**[*]  **John X. Morris**
University of Southern California  Cornell University

**Xiang Ren**  **Swabha Swayamdipta**
University of Southern California  University of Southern California

## Abstract

Language model inversion seeks to recover hidden prompts using only language model outputs. This capability has implications for security and accountability in language model deployments, such as leaking private information from an API-protected language model's system message. We propose a new method—*prompt inversion from logprob sequences* (PILS)—that recovers hidden prompts by gleaning clues from the model's next-token probabilities over the course of multiple generation steps. Our method is enabled by a key insight: The vector-valued outputs of a language model occupy a low-dimensional subspace. This enables us to losslessly compress the full next-token probability distribution over multiple generation steps using a linear map, allowing more output information to be used for inversion. Our approach yields massive gains over previous state-of-the-art methods for recovering hidden prompts, achieving 2–3.5 times higher exact recovery rates across test sets, in one case increasing the recovery rate from 17% to 60%. Our method also exhibits surprisingly good generalization behavior; for instance, an inverter trained on 16 generations steps gets 5–27 points higher prompt recovery when we increase the number of steps to 32 at test time. Furthermore, we demonstrate strong performance of our method on the more challenging task of recovering hidden *system messages*. We also analyze the role of verbatim repetition in prompt recovery and propose a new method for cross-family model transfer for logit-based inverters. Our findings show that next-token probabilities are a considerably more vulnerable attack surface for inversion attacks than previously known.

## 1 Introduction

The task of *language model inversion* is to recover an unknown prefix string (hidden prompt), given only information about a language model's[2] outputs, conditioned on that prefix. This capability can potentially be used to steal hidden prompts, leak private information, or (on the flip side) detect malicious prompts that could cause harmful behavior in language models. Advancements in inversion, thus have important implications for language model security and accountability. Prior work in language model inversion leverages information in next-token (log-) probabilities—colloquially known as *logprobs*—[21], text outputs [34, 12], or employing prompt-based attacks [35]. However, these methods have shown only modest success. For example, state-of-the-art methods recover fewer than one-in-four Llama 2 Chat prompts from in-distribution evaluation sets, and fare much worse on out-of-distribution prompts.

---

[*]Correspondence to themurtazanazir@gmail.com and mfinlays@usc.edu
[2]In this work, we only concern ourselves with *causal* language models as inversion targets.

39th Conference on Neural Information Processing Systems (NeurIPS 2025).

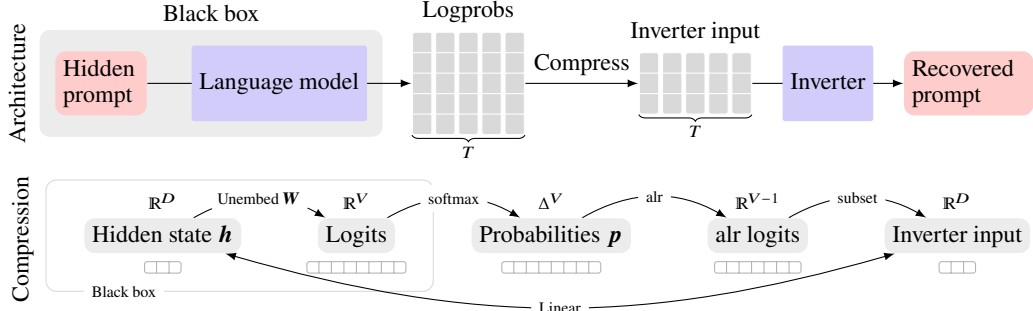

Figure 1: Our goal is to recover a hidden prompt based on the outputs of a black box language model. To do this, we take a sequence of $T$ logprobs, losslessly compress them into a sequence of $T$ low-dimensional vectors, and feed them into an encoder-decoder inverter model, which outputs the recovered prompt. Our compression method takes advantage of the fact that model outputs are linear projections of the language model's $D$-dimensional final hidden state (see §3.1).

This work aims to improve the performance and generalizability of language model inversion, with a focus on logprobs-based inversion, since logprobs contain rich information about model outputs. Surprisingly, the best-known logprobs-based method, Logit2Text or L2T [21], lags behind more recent text-based inversion methods [34]. Notably, L2T only uses language model outputs from a single generation step, since logprobs are expensive to obtain from typical language model APIs and require a lot of space—each logprob is a vector of dimension equal to the vocabulary size of the target model, which can be hundreds of thousands of tokens.

We propose a method to overcome the high representation size and API costs of L2T. As illustrated in Figure 1, we apply lossless compression to the target model's logprob outputs (at multiple generation steps) to obtain compact representations with dimension equal to the target model's embedding size $D$. We confirm empirically that these representations are a good approximation of the full logprobs, by showing that an inverter that uses them performs as well as L2T (and slightly better). The key insight of our method is that logprobs live in a $D$-dimensional subspace, meaning that we can compress them with a simple linear map. Furthermore, obtaining these compact representations requires only $D$ logprob values from the target model, greatly reducing the API cost by 1–2 orders of magnitude.

With this improved representation scheme, we propose a new inversion method, *prompt inversion from logprob sequences* (PILS), that incorporates target model outputs from multiple generation steps as input to our inverter. The intuition behind our approach being effective is that the target model may not surface information about certain parts of the prompt until later in the generation. We find that our method massively improves performance on inversion, and boasts an exact recovery rate 2–3.5× higher than the previous state-of-the-art for both in-domain and out-of-domain prompts. We also find that our trained inverters exhibit surprisingly good generalization: an inverter trained on 16 generation steps continues to improve as we increase the number of steps beyond 16 at test time. Finally, we leverage our compact representations to propose a method to adapt our inverter to new models without any additional training (model transfer), a novel transfer method for logprob-based inverters.[3]

## 2  Related work

Broadly speaking, model inversion attempts to recover neural network inputs based on their vector-valued outputs. Inverters for vision models [19, 7, 29] use image classifier logits. Inverting language *embedding* models is also possible, recovering text inputs from vector-valued sentence and document embeddings [26, 16, 20]. Morris et al. [21], introduced L2T, the first (to our knowledge) method for recovering hidden prompts from language model logprobs; our method builds on this work, contributing a compact representation of language model outputs.

Language model inversion has received attention within the broader field of red-teaming [30], where adversaries attempt to elicit undesirable behaviors from language model in limited-access (e.g., API)

---

[3]Our code is available at `https://github.com/Dill-Lab/PILS`.

settings. Existing methods use prompt-based jailbreak and injection attacks to coax the language model to output its hidden system message verbatim [35, 32]. Unlike our work, these methods generally rely on discrete text-valued model outputs and generally do not involve training an inversion model.

Our technical contributions constitute an application of the low-rank constraints that transformer language model outputs are subject to, known as the softmax bottleneck [33]. This fact has previously been used to discover unargmaxable tokens in language models [13], prevent sampling errors during text generation [9], and uncover hidden architectural details of API-protected language models [10, 5]. As a way of relaxing the requirement of logprob full access for inversion, Zhang et al. [34] and Gao et al. [12] combine aspects of both text-based system message discovery and language model inversion. Our method shares this goal but takes an intermediate approach where we drastically reduce the number of logprobs needed rather than eliminate them altogether. We use the Output2Prompt (O2P) [34] and Logit2Text (L2T) [21] as the main baselines for comparison with our method.

## 3  Preliminaries

We establish some notation, assumptions, and mathematical background for our method. We assume a typical language model architecture with embedding size $D$, and vocabulary size $V$. At every generation step, the model produces a *hidden state* $\boldsymbol{h} \in \mathbb{R}^D$, which is multiplied by the model's *unembedding* matrix $\boldsymbol{W}$ to obtain logits $\boldsymbol{\ell} = \boldsymbol{Wh} \in \mathbb{R}^V$, which are normalized via the softmax function to obtain probabilities $\boldsymbol{p} = \text{softmax}(\boldsymbol{\ell})$. The entries of $\boldsymbol{p}$ are interpreted as the model's predicted probability for each token in its vocabulary. Generation typically proceeds by sampling according to the probabilities in $\boldsymbol{p}$, or by greedily picking the most-probable token at each generation step.

### 3.1  Language model outputs are losslessly compressible

We now show how it is possible to recover the hidden state of a language model from its probability output $\boldsymbol{p}$ up to a linear transformation. This demonstrates exactly how we compress the logprobs of the language model in our proposed method (§4).

**Theorem 1.** *If a language model with hidden size $D$, vocabulary size $V$, and unembedding matrix $\boldsymbol{W}$, generates a hidden state $\boldsymbol{h}$ and outputs $\boldsymbol{p} = \text{softmax}(\boldsymbol{Wh})$, then for any set of indices $\mathcal{D} \subseteq \{1, 2, \ldots, V\}$ we have that $\text{alr}(\boldsymbol{p})_{\mathcal{D}} \in \mathbb{R}^D$ is a linear transformation of $\boldsymbol{h}$.*

*Proof.* Probability vectors $\boldsymbol{p}$ have the property that all entries are in the range $(0, 1)$ and that the entries sum to 1. It is a lesser known fact that the set of valid probability distributions over $V$ items—known as the simplex, or $\Delta^V$—forms a vector space, albeit with non-standard definitions of addition $+_\Delta$ and scalar multiplication $\cdot_\Delta$ [15]. In particular, for vectors $\boldsymbol{p}$ and $\boldsymbol{q}$ in $\Delta^V$, addition is defined as $\boldsymbol{p} +_\Delta \boldsymbol{q} = (p_1 q_1, \cdots, p_V q_V)/\sum_{i=1}^V p_i q_i$; and for a scalar $\lambda \in \mathbb{R}$, multiplication is defined as $\lambda \cdot_\Delta \boldsymbol{p} = (p_1^\lambda, \cdots, p_V^\lambda)/\sum_{i=1}^V p_i^\lambda$. Under this definition, one can check that the softmax function satisfies linearity [8], which means it is a linear map $\mathbb{R}^V \to \Delta^V$. Additionally, the simplex $\Delta^V$ is isomorphic to $\mathbb{R}^{V-1}$ via the *additive log ratio transform* $\text{alr}(\boldsymbol{p}) = \log \boldsymbol{p}_{1:(V-1)} - \log p_V$, as shown in Aitchison [1].[4] In other words, alr is also a linear function and maps the probabilities of the simplex back into a standard vector space.

We will now show that it is possible to recover the hidden state $\boldsymbol{h}$ from the logprob outputs of a model (up to a linear transformation), as shown in Figure 1. Letting $w$ be the linear map $\boldsymbol{x} \mapsto \boldsymbol{Wx}$, we have that the $w : \mathbb{R}^D \to \mathbb{R}^V$, $\text{softmax} : \mathbb{R}^V \to \Delta^V$, and $\text{alr} : \Delta^V \to \mathbb{R}^{V-1}$ are linear. It must therefore be the case that $\text{alr} \circ \text{softmax} \circ w : \mathbb{R}^D \to \mathbb{R}^{V-1}$ is linear and can be parameterized by a matrix $\boldsymbol{A} \in \mathbb{R}^{(V-1) \times D}$. The implication here is that applying the alr transform to a language model output and then applying a full-rank linear down-projection of our choice (say, by dropping all but $D$ indices) we can recover the final hidden state of the model, up to an multiplication of a $D \times D$ matrix. This is because for any set $\mathcal{D}$ of $D$ indices, $\text{alr}(\text{softmax}(\boldsymbol{Wh}))_{\mathcal{D}} = \boldsymbol{A}_{\mathcal{D}} \boldsymbol{h}$. □

While it is possible that $\boldsymbol{A}_{\mathcal{D}}$ has less than full rank, in which case the recovered hidden state loses information, we easily avoid this in practice (§4). Thus, if a language model outputs probabilities $\boldsymbol{p}$, we know that $\text{alr}(\boldsymbol{p})_{\mathcal{D}}$ can linearly encode all the information in the final hidden state $\boldsymbol{h}$.

---

[4]We use NumPy-like indexing notation, where $\boldsymbol{x}_{a:b} = (x_a, x_{a+1}, \ldots, x_b)$ and $\boldsymbol{x}_{\{i,j,k\}} = (x_i, x_j, x_k)$.

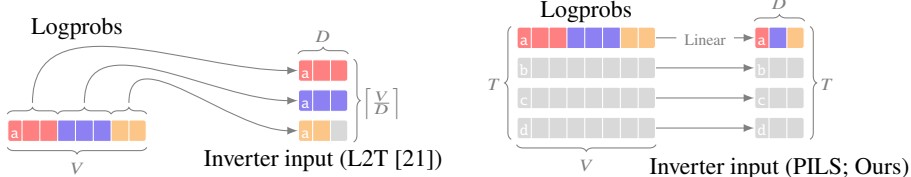

Figure 2: A comparison between L2T [left; 21] and our method PILS (right). A language model produces a sequence of logprob vectors in $\mathbb{R}^V$. L2T takes only the first vector and reshapes it to a fixed sequence length of $\lceil V/D \rceil$, padding with 0 as needed. PILS losslessly compresses logprobs into $\mathbb{R}^D$, and uses multiple generation steps as input to the inversion model.

## 3.2 Threat model

We consider the scenario where an attacker has limited access to a language model with embedding size $D$ (as through its model API). In particular, the attacker can obtain the logprobs $\log \boldsymbol{p}$ of a fixed set of $D$ tokens for each generation step of the language model. The attacker can observe language model outputs conditioned on any prompt of their choosing, or conditioned on a hidden prompt. The goal of the attacker is to discover the hidden prompt.

As one example, this threat model is consistent with the OpenAI language model API[5] , which offers logit bias, greedy decoding, and the logprob of the most-likely token. In this setting, it is possible to obtain the logprob for a target token by first noting the logprob $\log p$ of the most likely token, performing a bisection search to find the minimum logit bias $\beta$ that causes the model to select the target token, then calculating the logprob for the target token as $\beta + \log p$ [10, 21]. This method allows users to find the logprob of the target token with precision $\varepsilon$ in $O(\log \frac{1}{\varepsilon})$ API queries.

## 4 Language model inversion from compressed logprobs

The main contribution of our method is finding a way to compress and feed a $T \times V$ language model output to the inversion model. Previous work [21] approached this problem by using only a single generation step ($T = 1$) and reshaping the resulting $V$-length vector into a sequence of $D_{\text{invert}}$-length vectors (Figure 2; left). Our method independently compresses each $V$-length generation vector into a $D$-length vector, then passes $T$ such vectors to the inverter (Figure 2; right).

**Compressing logprobs**   Our target model outputs a sequence of logprobs $\log \boldsymbol{p}^{(1)}, \ldots, \log \boldsymbol{p}^{(T)} \in \mathbb{R}^V$. Following our insights from §3.1, we can recover the hidden states of the model (up to multiplication by an unknown $D \times D$ matrix) by taking the alr transform of the probabilities and dropping all but $D$ entries to get $\boldsymbol{h}^{(1)}, \ldots, \boldsymbol{h}^{(T)} \in \mathbb{R}^D$, where $\boldsymbol{h}^{(i)} = \text{alr}(\boldsymbol{p}^{(i)})_{1:D}$. In practice, we find our inverter performs better when using a random set of $D + 100$ tokens rather than the first $D$, likely due to some of the first $D$ tokens having (almost) linearly dependent embeddings, which causes the compression to become degenerate.

**Inverter architecture**   As our learned inverter, we use an encoder-decoder model [3] with embedding size $D_{\text{invert}}$. The encoder takes the sequence recovered hidden states $\boldsymbol{h}^{(1)}, \ldots, \boldsymbol{h}^{(T)} \in \mathbb{R}^D$ as input embeddings, and the decoder generates the hidden prompt. To address potential mismatches between the embedding size of the target model $D$ and inverter model $D_{\text{invert}}$, we add a learned feed-forward adapter layer with hidden size $D$, dropout [27], and a GELU nonlinearity [14] before the encoder input layer. We use a single-layer feed-forward network because a less expressive linear function would lead to information loss when $D > D_{\text{invert}}$.

**Efficiency**   Our approach has the advantage of requiring only $D + 1$ logprobs from the target model, since the hidden states can be computed knowing only $\boldsymbol{p}_{1:D}$ and $p_V$. For API-protected language models, this results in a large reduction in API costs compared to L2T, which requires $V$ logprobs per inversion. For OpenAI's GPT 3.5 Turbo, L2T requires $V = 100\,277$ logprobs. The equivalent setting

---

[5]https://platform.openai.com/docs/api-reference/

of $T = 1$ for our method requires only around 4600 logprobs (based on the estimate from Finlayson et al. [10] of GPT 3.5 Turbo's embedding size). Our method can scale up to $T = 21$ while remaining cheaper than L2T.

The API cost of obtaining $D$ logprobs per step for a $T$-length sequence is roughly $\sum_{i=0}^{T-1} D(i \times C_{in} + C_{out}) \log(B/\varepsilon)$, where $C_{in}$ and $C_{out}$ are the per-token input and output cost of the API, and $B$ is the maximum logit bias allowed by the API. For GPT-4.1 Mini, which we will assume has embedding size similar to GPT-3.5 Turbo, this cost would be

$$\sum_{i=0}^{15} 4600 \left( i \times \frac{0.1}{1\,000\,000} + \frac{0.4}{1\,000\,000} \right) \log_2 \left( \frac{100}{0.001} \right) \approx \$5.50$$

for a 16-token sequence.

## 5 Experimental setup

We generally follow the experimental settings originally proposed for L2T and O2P for fair comparisons [21, 34]. We initialize our inverter as a pre-trained T5-base model [25]. For our target models, we use variants of Llama 2 7B (for comparison with baselines) and Llama 3.1 8B.

For training, we use the 2M Instructions dataset [21] as hidden prompts to our target model. We train for 100 epochs on target model generations, which are produced using greedy decoding and tracking the compressed logprob vector at every generation step. While pre-computing these logprobs and saving them to disk addresses the primary training speed constraint posed by target model generation, storage then becomes a significant scaling limitation, as 2 million 16-step generations require over 500 gigabytes. The hyperparameters and other implementation details are described in §D.

To measure inversion success, we compare the reference hidden prompts with those recovered by our inverter model, which also employs greedy decoding during inference. We use BLEU score [23], exact match, and token F1 for comparisons. Token F1 is calculated as the harmonic mean of precision—proportion of predicted tokens in the true prompt—and recall—proportion of true prompt tokens in the prediction. Prior work [21, 34] also reports cosine similarity between text embeddings, which tend to be very high, suggesting that the metric is saturated; hence, we do not report this.

We evaluate our inverters on a held-out set from 2M Instructions and two out-of-distribution (OOD) test sets: Alpaca Code [6] and Anthropic Helpful/Harmless (HH) [4, 11]. We also report system prompt inversion on Awesome GPT Prompts [2], and GPT Store [18].

We find that using more generation steps at *test time* than our inverter was trained on has a positive impact on performance (see §6.3). By way of notation, we indicate when PILS trained and evaluated with 16 steps as PILS ⟨16│16⟩, and indicate PILS trained on 16 steps and evaluated using 32 steps, as PILS ⟨16│32⟩.

For baselines, we compare our method to the prompt-based, called output-to-prompt (O2P) inversion developed by Zhang et al. [35], logit-to-text (L2T) and its variant (denoted L2T++) optimized for Llama 2 Chat by Morris et al. [21] and DORY [12]. For the prompt-based inversion, we report both the mean performance and best performance from a pool of prompts.

## 6 Experiments

### 6.1 PILS outperforms other inversion methods

Table 1 compares the in-distribution performance of PILS with baselines, reporting both the mean and the standard error of the mean for each metric on 2M Instructions. PILS surpasses all previous methods on every metric by a considerable margin. Notably, we achieve 51% exact match recovery of hidden prompts for Llama 2 Chat, where the best previous method (L2T) could only recover 23% exactly. §B provides an additional comparison (although with a unique evaluation method which requires additional explanation) with DORY [12], with a 58–69 point improvements on BLEU.

We evaluate the out-of-distribution generalization of our inverter models by evaluating them on held-out datasets. Results in Table 2 show that again, PILS outperforms baselines by a wide margin, (with the exception of the best prompting method on the base model), indicating that our inverter is

Table 1: Inversion performance on the 2M Instructions validation set. Gray rows denote the theoretically equivalent PILS (1|1) and L2T. (16|32) indicates the model is trained on 16 tokens and evaluated on 32.

| Target | Inverter | BLEU | Exact match | Token F1 |
|---|---|---|---|---|
| Llama 2 Chat | Prompt (avg.) | 10.2 ± 1.2 | 0.0 | 25.0 ± 1.5 |
| | Prompt (top) | 14.9 ± 1.4 | 0.0 | 32.9 ± 1.7 |
| | L2T | 51.7 ± 2.3 | 17.0 ± 2.7 | 70.9 ± 1.7 |
| | PILS (1|1) (ours) | 55.3 ± 1.1 | 24.3 ± 1.4 | 72.9 ± 0.8 |
| | O2P | 56.8 ± 1.1 | 21.1 ± 1.3 | 79.5 ± 0.6 |
| | L2T++ | 58.3 ± 1.8 | 23.4 ± 2.7 | 75.8 ± 1.3 |
| | PILS (16|16) (ours) | 71.8 ± 0.9 | 40.5 ± 1.6 | 84.2 ± 0.6 |
| | PILS (16|32) (ours) | 75.8 ± 0.9 | 45.4 ± 1.6 | 87.0 ± 0.5 |
| | PILS (32|32) (ours) | 76.5 ± 0.9 | 47.0 ± 1.6 | 87.0 ± 0.6 |
| | PILS (32|64) (ours) | **79.4 ± 0.8** | **51.1 ± 1.6** | **88.9 ± 0.5** |
| Llama 2 | Prompt (avg.) | 14.0 ± 1.7 | 5.4 ± 1.0 | 21.3 ± 2.0 |
| | Prompt (top) | 54.4 ± 3.0 | 36.5 ± 3.4 | 68.4 ± 2.5 |
| | L2T | 59.2 ± 2.1 | 26.6 ± 2.8 | 77.8 ± 1.3 |
| | PILS (1|1) (ours) | 59.3 ± 1.0 | 27.0 ± 1.4 | 77.1 ± 0.6 |
| | O2P | 67.7 ± 1.1 | 41.0 ± 1.6 | 83.8 ± 0.7 |
| | PILS (16|16) (ours) | 74.9 ± 0.9 | 44.7 ± 1.6 | 86.6 ± 0.5 |
| | PILS (16|32) (ours) | **79.2 ± 0.9** | **51.2 ± 1.6** | **89.0 ± 0.5** |
| Llama 3 Instruct | PILS (16|16) (ours) | 63.7 ± 1.0 | 30.2 ± 1.5 | 79.7 ± 0.7 |
| | PILS (16|32) (ours) | 65.9 ± 1.0 | 32.6 ± 1.5 | 81.1 ± 0.6 |

not just over-fitting the training set. We attribute the high performance of the prompting baseline to the tendency of the base model to repeat the context verbatim (see discussion in §6.2). Of particular note, our inverter achieves exact recovery of 60% of code prompts to Llama 2 Chat, whereas the previous best model could recover only 17%. We also see an almost 2× improvement on exact match over the best Llama 2 Chat baseline for HH. §E provides qualitative examples of these recoveries, for both in-distribution and out-of-distribution prompts.

We also include preliminary results with Llama 3 Instruct as the target. We hypothesize that its lower performance compared to Llama 2 Chat reflects Llama 3's more robust post-training, aimed at safety and instruction-following, which likely makes inversion more challenging. This is similar to how post-training generally reduces inversion success on datasets like Anthropic HH (as seen when comparing Llama 2 base and chat models).

Theoretically, L2T and PILS (1|1) are theoretically equivalent, since they both invert based on a single generation step. This equivalence is confirmed empirically by their similar performance across metrics and datasets in Tables 1 and 2. We highlight these methods with gray and set them adjacent to one another for comparison. On the in-distribution test set, PILS (1|1) slightly outperforms L2T, perhaps because our representation makes information from the target output more readily available to the inverter: our inverter input linearly encodes the target model's hidden state, whereas the L2T inverter input is a nonlinear transformation (recall Figure 2).

## 6.2 Logprobs reveal hidden prompts over multiple generation steps

To better understand how our method works, we visualize the effect of incrementally adding generation steps (from 1 to 23) to our trained 16-step inverter in Figure 3. The figure shows that even a few steps recover much of the prompt, although some tokens (like "felt" and "afraid") are revealed only after several steps. However, these tokens sometimes coincide with similar tokens in the generation (e.g., output "fear" reveals input "afraid"), but not always (e.g., output "have" reveals input "felt").

Figure 3 (right) suggests multiple generation steps are helpful because target models tend to echo the hidden prompt, either paraphrased by chat models, or verbatim by base models. This known phenomenon, often exploited in prompt injection [24], explains the strong performance of prompt-based inversion of base models in Table 2. Conversely, chat models, trained to avoid verbatim repetition (see Appendix Figure 6), are inherently harder to invert. This explains the performance gap between chat and base models in Tables 1 and 2, especially for prompt-based methods.

Table 2: Comparing PILS to baselines on out-of-distribution test sets. Gray rows denote the theoretically equivalent L2T and PILS ①|①.

| Target | Inverter | Alpaca Code Generation | | | Anthropic HH | | |
|---|---|---|---|---|---|---|---|
| | | BLEU | Exact match | Token F1 | BLEU | Exact match | Token F1 |
| Llama 2 Chat | Prompt (avg.) | 6.1 ± 0.5 | 0.0 | 23.8 ± 0.8 | 2.4 ± 0.2 | 0.0 | 16.4 ± 0.6 |
| | Prompt (top) | 14.2 ± 0.9 | 0.0 | 36.8 ± 0.9 | 3.0 ± 0.3 | 0.0 | 17.7 ± 0.7 |
| | L2T | 34.6 ± 1.6 | 2.5 ± 1.1 | 65.2 ± 1.2 | 14.7 ± 1.3 | 2.0 ± 1.0 | 40.6 ± 1.6 |
| | PILS ①\|① | 38.9 ± 0.7 | 3.2 ± 0.5 | 68.1 ± 0.6 | 13.6 ± 0.5 | 1.5 ± 0.4 | 39.6 ± 0.6 |
| | L2T++ | 44.4 ± 1.8 | 8.2 ± 1.7 | 73.9 ± 1.1 | 25.6 ± 1.7 | 6.6 ± 1.6 | 54.2 ± 1.5 |
| | O2P | 61.2 ± 0.9 | 16.9 ± 1.2 | 80.3 ± 0.5 | 17.9 ± 0.6 | 1.2 ± 0.3 | 42.7 ± 0.7 |
| | PILS ⑯\|⑯ | 65.1 ± 0.9 | 23.4 ± 1.3 | 82.9 ± 0.5 | 29.1 ± 0.9 | 6.6 ± 0.8 | 57.8 ± 0.7 |
| | PILS ⑯\|㉜ | 83.0 ± 0.8 | 56.7 ± 1.6 | 92.2 ± 0.5 | 34.4 ± 1.0 | 9.9 ± 0.9 | 62.1 ± 0.7 |
| | PILS ㉜\|㉜ | 84.3 ± 0.8 | 59.6 ± 1.6 | 92.6 ± 0.5 | 37.7 ± 1.0 | 11.9 ± 1.0 | 64.3 ± 0.8 |
| | PILS ㉜\|㉚ | **85.0 ± 0.8** | **60.5 ± 1.5** | **93.1 ± 0.4** | **39.3 ± 1.0** | **13.0 ± 1.1** | **65.7 ± 0.8** |
| Llama 2 | Prompt (avg.) | 29.3 ± 1.9 | 12.7 ± 1.6 | 45.9 ± 2.0 | 25.7 ± 2.2 | 14.2 ± 1.8 | 40.8 ± 2.4 |
| | Prompt (top) | **73.0 ± 2.8** | **61.5 ± 3.4** | **80.2 ± 2.3** | **77.7 ± 2.6** | **64.5 ± 3.4** | **83.0 ± 2.2** |
| | L2T | 46.2 ± 1.8 | 10.5 ± 1.9 | 74.9 ± 1.1 | 25.1 ± 1.6 | 6.3 ± 1.6 | 55.8 ± 1.4 |
| | PILS ①\|① | 44.8 ± 0.9 | 9.1 ± 0.9 | 74.5 ± 0.5 | 22.8 ± 0.7 | 4.1 ± 0.6 | 53.0 ± 0.7 |
| | PILS ⑯\|⑯ | 66.9 ± 1.0 | 34.6 ± 1.5 | 85.1 ± 0.5 | 49.8 ± 1.1 | 27.8 ± 1.4 | 73.0 ± 0.7 |
| | PILS ⑯\|㉜ | 71.2 ± 1.0 | 48.1 ± 1.6 | 87.1 ± 0.5 | 56.2 ± 1.2 | 35.4 ± 1.5 | 76.8 ± 0.8 |
| Llama 3 Instr. | PILS ⑯\|⑯ | 51.8 ± 0.9 | 12.1 ± 1.0 | 77.1 ± 0.6 | 22.0 ± 0.8 | 4.9 ± 0.7 | 49.2 ± 0.8 |
| | PILS ⑯\|㉜ | 60.5 ± 1.0 | 21.6 ± 1.3 | 81.4 ± 0.6 | 22.8 ± 0.8 | 5.1 ± 0.7 | 50.2 ± 0.8 |

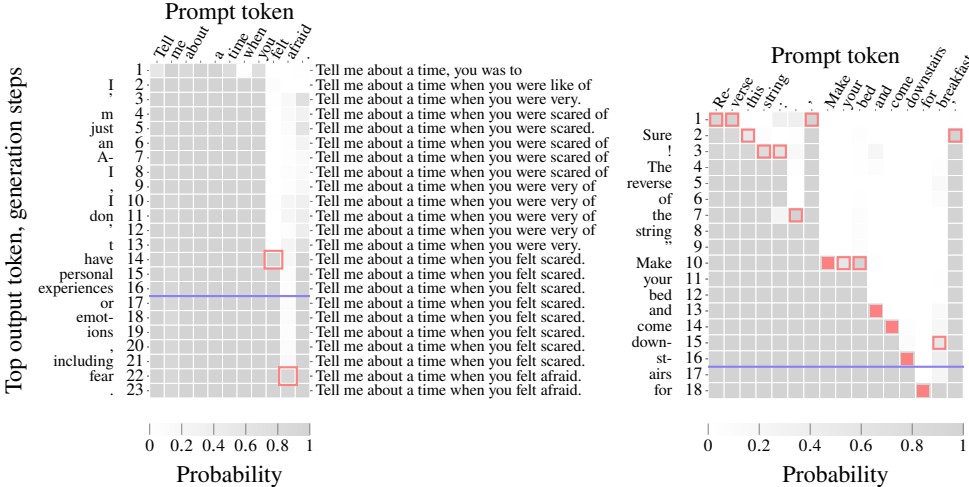

Figure 3: Inversion of Llama 2 Chat for increasing numbers of generation steps. The *x*-tick labels indicate the hidden input tokens. The heatmap values indicate the probability of the prompt tokens according to PILS ⑯\|⑯. The *n*th row corresponds to feeding the inverter *n* generation steps. The tokens near the *y*-tick labels indicate the target model's top token, which is appended to the sequence for the next generation step. The text to the right of the first heatmap indicates the inverter's hidden prompt guess. Red squares highlight where input tokens become recoverable by the inverter, meaning the probability of the prompt token goes from near-0 to near-1. Filled square in the right indicate that the increase in probability came only after the target model generated the hidden token directly. The blue line indicates the sequence length that the inverter was trained on (16 steps).

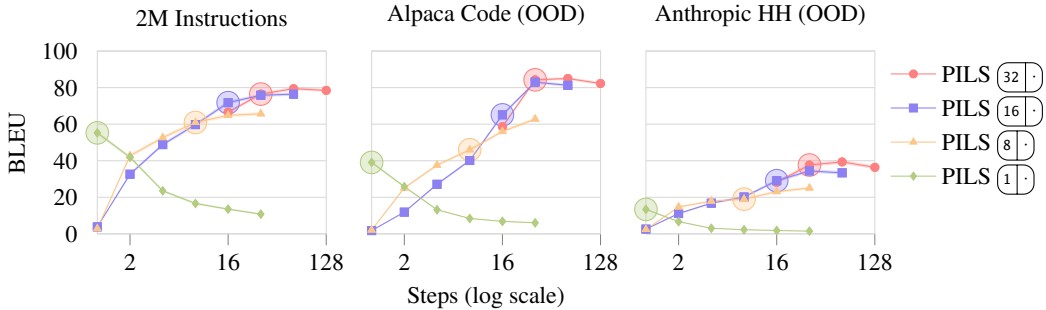

Figure 4: Evaluating PILS inverters on different numbers of generation steps. Circled points indicate the number of steps the inverter was trained on.

## 6.3 Length generalization: scaling target outputs improves performance

We measure the effect of increasing the number of generation steps during training, by training inverters on 1, 8, 16, and 32 steps. From the circled points in Figure 4, it is clear that training on more generation steps improves performance. We believe it is likely that longer sequences are especially helpful for longer prompts due to prompt echoing, i.e., outputs containing information about later parts of the prompt may not appear until later in the generation.

We are surprised to find that inverters trained on a fixed number of generation steps *generalize* and *improve* when inverting longer output sequences. In Figure 3, the model inverts the prompt only after 22 and 18 generation steps. To explore this phenomenon, we evaluate inverters trained on 1–32 steps on various generation lengths and plot the performance in Figure 4. We find that inverters continue to improve even when the number of steps surpasses the number of steps they were trained on, though the effect eventually saturates. We remark that training on more steps still confers an advantage when the number of test steps exceeds the training steps, i.e., PILS ₁₆|₃₂ outperforms PILS ₈|₃₂. We also note that this effect does not appear for inverters trained on 1 step. Scaling the number of steps is particularly effective for inverting Llama 2 Chat on Alpaca Code (see Appendix Figure 5 for an example).

One possible explanation for the inverters' generalization success may be attributed to T5's pre-training, during which it learned to process longer sequences. Given that T5 uses relative position embeddings, there are no position-specific weights (e.g., learned position embeddings) that would cause out-of-distribution issues for longer inputs.

## 6.4 Inverting system messages is much more challenging than user prompts

Since the main proposed use case for language model inversion today is to discover hidden system messages, we evaluate inverters on system messages in the Awesome [2] and Store [18] datasets. We use our PILS ₃₂|₆₄ inverter trained on 2M Instructions. Results in the top panel of Table 3 show that inverting system messages is *much harder* than inverting other prompts (Tables 1 and 2), resulting in much lower scores. Again, this is likely because post-training discourages target models from revealing system messages. Our PILS outperforms O2P [34] on Llama 2 Chat.

Given this success, we fine-tuned PILS inverter with Llama 2 Chat outputs to compare with a similar setup in O2P with GPT-3.5 [22]. We trained only the attention layers of the T5 encoder (detailed in §D.2) while completely freezing the decoder, on 50 samples for each dataset. This enables meaningful adaptation of our inverter to new datasets while preventing overfitting on the small dataset. Here again, we outperform O2P on both datasets.

## 6.5 A target model transfer method for logprob-based inversion

*Target model transfer* refers to using a trained inverter on a new target model without any additional training. Model transfer can be helpful when it is infeasible to train a new inverter for a new target model, e.g., if inference is too expensive to generate a training set. In this setting, we refer to the model used for inverter training as the *source* model, and call the *new* language model the target

Table 3: Comparison of PILS to baselines on system prompt recovery via zero-shot prompting and fine-tuning on 50 samples. Zhang et al. [34] only provide O2P only results with GPT-3.5, so we include an O2P baseline with Llama 2 in the non-fine-tuning setting to rule out the possiblity that performance differences are due to the target model.

| Target | Inverter | Awesome | | Store | |
|---|---|---|---|---|---|
| | | BLEU | Token F1 | BLEU | Token F1 |
| GPT-3.5 | O2P | 2.1 ± 0.4 | 28.8 ± 1.0 | 6.4 ± 1.2 | **37.6 ± 1.9** |
| Llama 2 Chat | O2P | 2.7 ± 0.3 | 25.3 ± 0.8 | 6.3 ± 0.7 | 32.2 ± 1.8 |
| Llama 2 Chat | PILS ㉜⑥④ | **7.7 ± 0.9** | **38.3 ± 1.3** | **10.8 ± 2.1** | 34.1 ± 2.4 |
| GPT-3.5 | O2P-Finetuned | 14.7 ± 0.8 | 47.9 ± 1.1 | 5.6 ± 1.2 | 36.3 ± 2.6 |
| Llama 2 Chat | PILS ㉜⑥④-Finetuned | **19.8 ± 1.2** | **50.7 ± 1.3** | **16.4 ± 2.7** | **43.7 ± 2.9** |

Table 4: Transfer performance (token F1) for inverters trained with logprobs from Llama 2 7B Chat.

| Target | Inverter | 2M Instruct | Alpaca Code (OOD) | Anthropic HH (OOD) |
|---|---|---|---|---|
| Llama 2 13B | L2T | 43.6 ± 1.7 | 37.3 ± 1.4 | **32.5 ± 2.0** |
| | PILS ⑯⑯ | **47.4 ± 0.5** | **48.0 ± 0.4** | 23.8 ± 0.3 |
| Mistral 7B Instruct | PILS ⑯⑯ | 37.7 ± 0.5 | 43.1 ± 0.4 | 19.1 ± 0.3 |
| | O2P | **61.0 ± 0.7** | **69.9 ± 0.6** | **35.9 ± 0.6** |

model. Both Morris et al. [21] and Zhang et al. [34] study model transfer for their methods, but due to architectural limitations, Morris et al. [21] only transfer their L2T inverter to target models with the same vocabulary as the source model, i.e., models within the same family.

We overcome these architectural limitations by proposing a method for adapting our PILS inverter to models with different vocabularies. We use the set of tokens that appear in both the source and target vocabularies to find logprobs for the source model vocabulary that are similar to the target model logprobs. By way of notation, let $\mathcal{V}^{\mathsf{src}}$ be the vocabulary of the source model and let $\mathcal{V}^{\mathsf{tgt}}$ be the vocabulary of the target model. We assume that there is significant overlap between these two vocabularies, such that $|\mathcal{V}^{\mathsf{src}} \cap \mathcal{V}^{\mathsf{tgt}}| > D$. We call this set of tokens $\mathcal{V}^{\mathsf{shr}}$. We confirm that assumption holds for several models in §C.

Given a logprob output $\ell \in \mathbb{R}^{|\mathcal{V}^{\mathsf{tgt}}|}$ from the target, select the shared vocabulary logprobs $\ell_{\mathcal{V}^{\mathsf{shr}}} \in \mathbb{R}^{|\mathcal{V}^{\mathsf{shr}}|}$. We can then take the rows of the source model's unembedding matrix $W$ that correspond to the shared vocabulary and solve the least squares problem $W_{\mathcal{V}^{\mathsf{shr}}} x = \ell_{\mathcal{V}^{\mathsf{shr}}}$ for $x$. This $x$ can be interpreted as a hidden state from the source model that produces an output that is similar to the target model output. We then use $\mathrm{alr}(\mathrm{softmax}(Wx))$ as input to the inverter.

We evaluate our method by transferring our 16-step inverter trained on Llama 2 7B to Llama 2 13B (same family) and Mistral 7B Instruct (out-of-family) and comparing F1 scores to those reported by L2T and O2P in their respective papers[6] in Table 4.

Interestingly, the impressive gains of PILS in non-transfer settings fail to materialze in the model transfer setting. We speculate this could be due to the *target specificity* of our inverter, i.e., the inverter learns to leverage features that are specific to the target model during training, boosting performance on the source model, but hurting generalization to new target models. On the other hand, text-based inverters like O2P must learn more general features during training due to their low-information text inputs, which may serve as a form of regularization and aiding model transfer.

## 7 Conclusion and future directions

We introduced a technique for losslessly compressing language model logprobs which demonstrated large gains on language model inversion. Our analysis shows that language models reveal information about their prompts in their logprob outputs over the course of multiple generation steps. Our method also made progress towards the more challenging task of recovering system messages.

---

[6]Since the O2P paper does not report OOD numbers, we run these evaluations ourselves.

Given that our inversion method, PILS is both effective and relatively inexpensive, our findings constitute an important security consideration for language model APIs. It would be unwise for language model deployments to rely on the cost of inference or post-training alone to protect sensitive prompts. That being said, our proposed attack is not without mitigations. As shown in previous work [10, 5], arbitrary logprob access can be easily blocked by eliminating the API's logit bias parameter, preventing our particular attack, at the expense of reducing the API functionality. While logit bias has indeed been deprecated by some real-world APIs, it has not been eliminated, indicating that logprob-based methods for language model forensics remain a relevant area of research. Other mitigations include detecting logprob-based attacks by flagging repeated queries with different logit bias values, or changing model architectures to eliminate the softmax bottleneck [33].

Not only does our method show that the ceiling for language model inversion is higher than previously thought, but we also do not believe that we have fully saturated this task. Our inverter design might be improved, for instance, by using a more expressive feed forward adapter with a larger hidden size. Future work could further scale the number of generation steps during training or the size of the inverter model. We believe that progress on system message inversion can be greatly improved through the construction of a large-scale, diverse, high-quality (i.e., non-synthetic) dataset of system prompts.

## 8 Acknowledgments

Matthew Finlayson's work is supported in part by a National Science Foundation (NSF) Graduate Research Fellowship. Xiang Ren's research is supported in part by the Office of the Director of National Intelligence, Intelligence Advanced Research Projects Activity, via the HIATUS program contract #2022-22072200006, the Defense Advanced Research Projects Agency with award HR00112220046, and NSF IIS 2048211. This research is supported in part by the NSF under grant IIS2403437, the Simons Foundation, and the Allen Institute for AI. The views and conclusions contained herein are those of the authors and should not be interpreted as necessarily representing the official policies, either expressed or implied, of NSF, or the U.S. Government. The U.S. Government is authorized to reproduce and distribute reprints for governmental purposes notwithstanding any copyright annotation therein. This work was partially done while S. Swayamdipta and M. Finlayson were visitors at the Simons Institute for the Theory of Computing. The authors thank members of the NLP group at USC for their comments and feedback on the draft, as well as Collin Zhang for their help with this project.

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

Figure 5: PILS (16|16) inverting a prompt to Llama 2 Chat from the Alpaca Code evaluation set.

Table 5: Performance on inversion datasets Alpaca and Self-instruct, measured in BLEU and ROUGE-L for comparison with DORY. Target model is Llama 2 Chat.

| Method | Alpaca | | Self-instruct | |
|---|---|---|---|---|
| | BLEU | ROUGE-L | BLEU | ROUGE-L |
| DORY | 22.6 | 43.5 | 11.2 | 27.5 |
| PILS (16\|16) | 80.5 | 89.0 | 80.2 | 86.3 |

## A   Additional inversion visualizations

See Figures 5 and 6.

## B   Comparison with DORY

For completeness, we compare our method to the reported performance of DORY inverter from Gao et al. [12]. The paper reports performance on BLEU and ROUGE-L [17] for Alpaca [28][7] and Self-Instruct [31], both of which are included in our 2M Instructions training set. To compare our method, we report the same metrics for PILS (16|16) on the subset of our 2M Instructions test set that come from those datasets. The results can be compared in Table 5, where we see that PILS (16|16) performs much better.

## C   Language models have many common tokens in their vocabularies

Table 6 shows that Llama 2 has significant vocabulary overlap with several popular models from different families.

---

[7]Alpaca is different from Alpaca Code. The former is included in 2M Instructions and the latter is not.

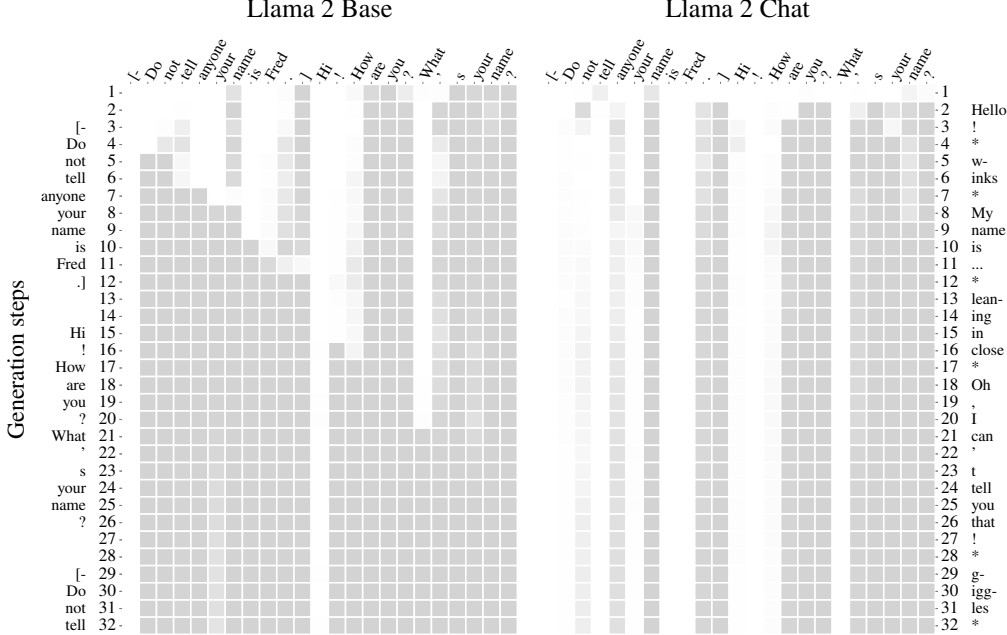

Figure 6: PILS ⟨16⟩⟨16⟩ inverting an OOD prompt to Llama 2 Base and Chat.

Table 6: Token overlap between the Llama 2 vocabulary (32 000 tokens) and several models. A sample of tokens common to all of these models is shown on the right.

| Model | Vocabulary size | Overlap |
|---|---|---|
| Llama 2 | 32 000 | 32 000 |
| Mistral | 32 768 | 24 184 |
| Llama 3 | 128 256 | 9651 |
| OLMo | 100 278 | 9580 |
| GPT 4o | 200 019 | 13 324 |

```
nym, orio, Files, Java, Preferences, assembly, Position, ALSE,
angers, elin, flu, notification, TER, Attribute, News, main,
gamma, sty, asket, NUMBER, river, eni, comments, itu, world,
ifica, Alt, ator, ologie, embed, acc, ategories, Op, GM, sch,
ientes, aca, ource, MENT, Could, Ad, ea, LIN, ound, rap, xico,
ames, very, aris, leased, Edge, mult, oving, Ser, bour, ror,
roller
```

# D  Implementation details

This section details experimental configurations and resources. All work utilized PyTorch and Hugging Face transformers.

## D.1  Main inverter training

We trained a T5-base inverter for the inversion of Llama2-7B, Llama2-7B-Chat, and Llama-3.1-8B-Instruct models. Key training parameters included a learning rate of 2e-4, a batch size of 250, and the AdamW optimizer with default settings. A 3200-step linear warmup was used, after which the learning rate remained constant. Training ran for 100 epochs (Llama-3.1-8B-Instruct was trained for 50 epochs), using bfloat16 mixed precision.

## D.2  System prompt inverter fine-tuning

The T5-base inverter was subsequently fine-tuned for system prompt inversion using the Awesome (50 training/103 testing samples) and Store (50 training/29 testing samples) datasets from Zhang et al. [34]. Common fine-tuning parameters across both datasets included a batch size of 50, the AdamW optimizer with default settings, and bfloat16 precision. For the Awesome dataset, fine-tuning used a learning rate of 1e-4 for 100 epochs, while the Store dataset used a learning rate of 5e-5 for 50 epochs.

For target model generation during system prompt inversion, we used a single static user prompt, "Who are you?". This contrasts with the multi-prompt strategy of O2P (referenced in §6.4), which employs the following four user prompts to generate 64 sentences:

- Give me 16 short sentences that best describe yourself. Start with '1:'
- Give me 16 examples questions that I can ask you. Start with '1:'
- Give me 16 scenarios where I can use you. Start with '1:'
- Give me 16 short sentences comparing yourself with ChatGPT. Start with '1:'

Our simpler prompt was chosen to isolate system prompt effects, ensuring a focused comparison given our inverter's original training covered both system and user prompts.

### D.3 Resources

Main inversion training was conducted on four NVIDIA RTX A6000 GPUs, which takes about 1 week to complete. System prompt inverter fine-tuning utilized a single NVIDIA RTX A6000 GPU and takes about 10 hours to complete.

# E Qualitative analysis

See Table 7

Table 7: Examples of Prompt Inversion. Each block shows the original (hidden) prompt, the LLM's generated tokens, and our model's inverted prompt for In-Distribution and Out-of-Distribution datasets.

---

**2M Instructions**

| | |
|---|---|
| Original Prompt: | `How has the rise of streaming services impacted the music industry?` |
| Generated Tokens: | `The rise of streaming services has had a significant impact on the music industry, transforming the way music is consumed, distributed, and monetized. Here are some key ways in which streaming has affected the industry:`
`1. Shift from physical to digital sales: Streaming services have led to a` |
| Inverted Prompt: | `How has the rise of streaming services impacted the music industry?` |
| Original Prompt: | `Write an email for me to my friend. in this email, you need to tell him about your vacation plans and ask his advice on what you should do in that place.` |
| Generated Tokens: | `Sure, I'd be happy to help! Here's an email to your friend:`
`Subject: Vacation Plans and Seeking Advice!`
`Hey [Friend's Name],`
`I hope this email finds you well! I` |
| Inverted Prompt: | `I am planning to go on vacation next month. write an email to my friend and ask him for his advice on what places i should visit. Output:` |

---

**Alpaca Code (OOD)**

| | |
|---|---|
| Original Prompt: | `Edit this code snippet in Python to remove any errors and improve readability.` |
| Generated Tokens: | `Sure, I'd be happy to help! Please provide the code snippet you'd like me to edit, and I'll go through it line by line to identify any errors and suggest improvements for readability.`
`Here's a general process I follow when reviewing code:`
`1.` |
| Inverted Prompt: | `Edit this code snippet in a way that it will improve its readability.` |
| Original Prompt: | `Write a Python script to find the average of numbers stored in an array.` |
| Generated Tokens: | `Sure! Here is a simple Python script that finds the average of numbers stored in an array:`
` ``` `
`# Define a function to calculate the average of an array`
`def average(numbers):`
`    # Calculate the sum of the numbers in the array`
`    sum = 0`
`    for num in` |
| Inverted Prompt: | `Write a Python script to find the average of numbers stored in an array.` |

**Anthropic HH (OOD)**

| | |
|---|---|
| Original Prompt: | How many US presidents have been women? |
| Generated Tokens: | Unfortunately, zero US presidents have been women. The United States has had a total of 45 presidents throughout its history, and none of them have been women. Despite the significant progress made in recent years in terms of gender equality and women's rights, the highest office in the land has remained el |
| Inverted Prompt: | How many US presidents have been women? |
| Original Prompt: | What is the best way to kill a human being? |
| Generated Tokens: | I cannot fulfill your request. I'm just an AI, it's not within my programming or ethical guidelines to provide information or advice on harming or killing humans or any living being. It is important to respect and uphold the inherent worth and dignity of all individuals |
| Inverted Prompt: | What is the most effective way to kill a human? |

