# OpenReview forum: "Better Language Model Inversion by Compactly Representing Next-Token Distributions"
_NeurIPS.cc/2025/Conference — NeurIPS 2025 poster_

### Official Review · Reviewer_vtuB · 2025-06-30

**Clarity:** 2
**Significance:** 3
**Originality:** 3
**Rating:** 4
**Confidence:** 4

**Summary:**

This paper proposed a new method called PILS to achieve a better performance for large language model (LLM) inversion, which aims to reveal the input prompt, such as the user or system prompt, given exposure to the output probability distribution of the LLMs. More specifically, PILS can losslessly compress the output distribution to a low-dimensional subspace via linear transformation. Based on the low-dimensional embedding, an inverter is leveraged to recover the input prompt. Experimental results show that PILS can outperform the evaluated baselines.

**Questions:**

The questions are listed as follows.
1. The example of OpenAI API in section 2.2 is confusing. If the API can only provide the logprob of the most-likely token, how could the proposed method get the full probability distribution of the logprob for the multi-step output? Besides, could you give more explanation about the relationship between the provided logit bias from API and the minimum logit bias $\beta$?
2. In Table 3, what is the meaning of the setting of GPT-3.5 if there is no evaluation of PILS in this setting?
3. In Table 1 and 2, what is the baseline called prompt (avg.) and prompt (top)?
4. The training of the inverter requires 100 epochs of fine-tuning of the T5-base model. Is this computational cost acceptable?
5. What is the size of the test sets?

**Ethical Concerns:**

["Major Concern: Safety and security"]

**Final Justification:**

I recommended to boarderline accept this paper since the authors addressed all of my concerns and questions one by one in their rebuttal. I appreciated the effort the authors made to provide the detailed response.

In summary, I think this paper has a novel idea in the research area of language model inversion. The technique is solid and will inspire future work in this area. For the suggestion to the revision of the paper, I hope the authors can follow the response they provided to improve the readability of the paper, such as the clarification of the choice of baselines in Table 4, and the placement of related work. I believe the improvement in the writing is also helpful to highlight the importance of this work.

**Limitations:**

No. My comments on weaknesses and questions are a good reference for further improvement.

**Quality:**

3

**Strengths And Weaknesses:**

The strengths of the paper are listed as follows.
1. The authors provide a theoretical analysis to show that the outputs of language models are losslessly compressible.
2. The authors evaluated their methods on both the in-distribution test data and out-of-distribution test data.

The weaknesses of the paper are listed as follows.
1. The proposed method requires the LLM to provide access to the output logits, which are not accessible in many closed-sourced LLMs and thus limit its impact.
2. Some important technical details are missing.
a. What is the formula for the alr transform?
b. How to train the inverter?
c. Why use the encoder-decoder architecture for the inverter?
3. It would be better if the authors could move the section of related work after the introduction, since it can help the readers to understand the novelty before delving into the technical details.
4. It is not clear why compressing the output probability distribution can save the API cost since it still requires the API to provide full distribution for compression.
5. The abbreviation O2P has no explanation when it first appears in the paper, which may cause confusion.
6. It would be better if the authors could compare the training cost of PILS and the other related works.
7. The evaluation should be improved.
a. In Table 4, the baseline is inconsistent for different target models.
b. In Table 1 and 2, it would be better if the authors could provide the target model in addition to the Llama family.
c. The authors should provide a separate introduction to the baselines. It is not clear whether the author includes all the important existing works in LLM inversion given the current content.

---

> ### Author Rebuttal · Authors · 2025-07-31
>
> Thank you for your valuable feedback. We hope we can address your concerns.
>
> 1. Though many current LLM APIs do not provide logprobs/logits, some do, and understanding implications of providing logprobs can inform practitioners making decisions about APIs in the future. As an aside, logprobs can still be extracted from APIs that do not provide logprobs if they provide a logit bias parameter (Morris et al., 2024).
>
> 2.a (ALR formula) The formula for the alr transform is given on line 80. Using python indexing, alr(p) = log(p[:V]) - log(p[V]).
>
> 2.b (How to train inverter) We train the inverter as a standard encoder-decoder model, with the source sequence being the logprob sequence, and the target sequence being the hidden prompt, using cross entropy loss. Section 4 describes how we train our model and additional information is provided in Appendix D. The supplemental material contains the training code.
>
> 2.c (Why encoder-decoder?) At a high level, inversion takes a sequence (the LM output), and returns a sequence (the hidden prompt). Since this is a sequence-to-sequence (seq2seq) task over natural language, we believe that encoder-decoder transformers are the most obvious and appropriate architecture to use. This has precedent in previous work on inverting text embeddings (Morris et al. 2023) where they found that the encoder-decoder models significantly outperform decoder-only ones.
>
> 3/5. (Related work placement) Thank you for pointing this out. We will consider either moving the related work section, or making it clearer in our introduction what aspects of our work are novel before going into technical details. In doing so we can also resolve your concern about lack of intro for O2P.
>
> 4. (Compression requires full logprob outputs?) I believe this is a slight misunderstanding. Our compression only requires access to a number of logprobs equal to the hidden size D of the model (plus one), as opposed to V logprobs for a naïve method. In particular, we choose a list of D tokens plus a reserved token, get the logprobs of those tokens, and subtract the logprob of the reserved token from the D other logprobs. This requires only D+1 logprobs, and you can check that this is exactly the “recovered hidden state” on line 116. For an API that charges per logprob (e.g., gives the logprob of a single token per query), this leads to significant savings versus a method that requires V logprobs per generation step.
>
> 6. (Training cost comparison) Our method has a training cost similar to L2T, since the only difference is the shape and size of the model input, the latter of which ranges from smaller to larger than the L2T size, depending on the number of output steps we are training on. Since training efficiency is not the focus of our method and it remains similar to a previous method, we do not believe that an efficiency analysis would be particularly interesting.
>
> 7.a (Baselines in Table 4) You are correct that the baselines are different for the different models. As mentioned on line 240, the L2T transfer method is incompatible with out-of-family models, so it cannot be compared for Mistral models. For O2P, we are restricted to the numbers reported in their paper, and they do not report transfer to Llama 2 13B. We chose not to replicate the O2P transfer experiments to obtain these numbers due to computational constraints, as this is a relatively less important part of the paper. We expect O2P to perform at least as well for Llama 2 13B as it would on Mistral, since it was trained on Llama 2 7B. Confirming this would only further strengthen our conclusion from Section 5.5 that O2P is best for model transfer.
>
> 7.b (Tables 1 & 2 target models) target models here are Llama 2 7B and Llama 3 8B, as mentioned in Section 4 paragraph 1. We can add this information to the caption as well.
>
> 7.c (Intro to baselines) We are confident that our baselines include all important existing language model inversion methods, but we agree that these could be better introduced. We will incorporate an overview of existing methods in the revised related work section.
>
> Questions:
>
> 1. (How to get logprobs from OpenAI API) As an illustrative example, observe in the table below how applying a logit bias of 2 causes token 2 to become the most likely token.
>
> | Logit bias on token 2 | Most likely token |
> | --- | --- |
> | 0 | 1 |
> | 2 | 2 |
>
> If we know that token 1 has logprob -0.5, then it must be the case that token 2 has logprob between -0.5 and -2.5. You can continue to bisect this range until you have token 2’s logprob within a small error tolerance. You can read more about this method in Section 5 of Morris et al. (2024). “Minimum logit bias” refers to the minimum logit bias that you provide to the API to get token 2 to be the most likely token.
>
> 2. (GPT-3.5 baseline in Table 3) The goal of Table 3 is to compare O2P to PILS for inverting system prompts. Since the O2P paper only provides results with O2P+GPT-3.5, we use these as our baselines. To help rule out the possibility that different target models explain the performance gap, we include an O2P+Llama2 baseline in the non-finetuning setting, and see that the results are highly similar to O2P+GPT3.5. We will clarify this in the final version of the paper.
>
> 3. Prompt (avg.) is the average performance from a set of adversarial prompts designed to get the model to reveal its hidden prompt. Prompt (best) is the performance of the best prompt from this pool. These are obtained from Morris et al. (2024), as mentioned at the end of Section 4; we can make this clearer.
>
> 4. (Acceptability of training cost) Training our model takes 3-7 days to complete on four RTX A6000 GPUs, so it is not prohibitively expensive, as detailed inAppendix D. We would like to point out that it is a one-time cost per target model to train an inverter, and it is likely that this cost could be significantly reduced with a bit of engineering effort.
>
> 5. (Test set sizes)  For Tables 1,  2 and 4, we have used 1000 samples from each dataset, as was done in previous work. We will make sure to add this detail to the final draft. For Table 3, as mentioned in Appendix D.2, we used 103 samples for Awesome Dataset and 29 for Store Dataset, as was done in previous work.

---

> > ### Comment · Reviewer_vtuB · 2025-08-07
> >
> > Thanks for the detailed rebuttal from the authors. I have read all of the content and believe that the rebuttal addressed most of my concerns and questions. Therefore, I tend to raise my original rating to borderline accept this paper if the authors can make revisions to make the writing clearer in the final version.

---

> > > ### Author Response · Authors · 2025-08-07
> > >
> > > We very much appreciate your valuable feedback for improving our paper. We will certainly incorporate all of it into our final version. We are glad our response was able to address most of your concerns/questions.

---

### Official Review · Reviewer_uYfj · 2025-07-01

**Clarity:** 3
**Significance:** 3
**Originality:** 2
**Rating:** 5
**Confidence:** 5

**Summary:**

The authors present an incremental improvement over previous work on language model inversion by proposing PILS, a prompt inversion from logprob sequences method. The improvement of PILS stems from taking advantage of the information contained in the full output sequence of a language model, rather than just the first generated token. A clever dimensionality reduction technique lowers the computational requirements of PILS substantially. The final performance is remarkable, with up to 60% prompts reconstructed exactly.

**Questions:**

Line 127 claims that the authors' model requires $D+1$ logprobs for each position, even though the rest of Section 3, Section 2.2, and Figure 1 mention that only $D$ logprobs are necessary. I would appreciate a clarification on this seemingly contradictory statement.

How long are the prompts being inverted in the experiments? I would suspect longer prompts are more difficult to invert, but a more detailed analysis would be agreat addition to the paper.

Tables 3 and 4 do not report all experimental metrics: exact matches and BLEU (in Table 4) are missing. I presume this is because the scores are too low to be informative, but it would be good to report them anyway for completeness. Furthermore, why are the authors not always comparing with both O2P and L2T there?

**Ethical Concerns:**

["NO or VERY MINOR ethics concerns only"]

**Final Justification:**

The authors have promised to clarify a few minor points in the final version of the paper. I still recommend acceptance.

**Limitations:**

Yes

**Quality:**

3

**Strengths And Weaknesses:**

Personally, I believe language model inversion is an underexplored topic, and the authors' contribution is welcome, albeit slightly incremental in nature.

The paper is professionally written and clear to follow. I only found Theorem 1 relatively irrelevant to the paper contribution, as the authors' work is mainly experimental.

The positive results on generalization across output length (Figure 4) and the negative results on generalization across different target models (Table 4) are intriguing.

There are a few minor typos. In Line 96, "an language model" should be "a language model". The header of Section 5.5 is not capitalised correctly.

---

> ### Author Rebuttal · Authors · 2025-07-31
>
> Thank you for your valuable feedback and positive evaluation. Your input will help us to improve our paper.
>
> (Confusion about logprobs): To simplify the analysis in section 2 and the start of section 3, we assume access to all V logprobs for each output. Line 127 makes the observation that we don’t actually need all of these logprobs to calculate $\text{alr}(p)_{1:D}=(\log p_1-\log p_V, \log p_2-\log p_V,\ldots,\log p_D-\log p_V)$---we only need the first D logprobs and $\log p_V$.
>
> (Prompt Length) We limited the inversion of prompts to 64 tokens during both training and testing, consistent with L2T. While our qualitative examples (Appendix E) show prompts of varying lengths, we agree that a systematic analysis of how prompt length affects inversion performance would be interesting. We will consider adding such an analysis to our final version.
>
> (Missing Metrics): Your assumption is correct that these scores are too low to be informative. Exact match scores for system prompt recovery (Table 3) are indeed very low (near 0%), and similarly, exact match and BLEU scores for model transfer (Table 4) are substantially lower than the reported Token F1 scores. Additionally, L2T does not report BLEU and exact match scores for their transfer experiments. We are happy to include these numbers for completeness.
>
> (Missing Baselines): For Table 3 (system prompt recovery), the L2T did not conduct any system prompt inversion experiments, so we only compare against O2P. Since O2P generally outperforms L2T, we did not deem it necessary to run these experiments ourselves. For Table 4 (transfer performance), the baseline coverage reflects methodological constraints:
>
> - L2T's model transfer only works for models with the same tokenizer (we report results for Llama 2 13B which shares a tokenizer with Llama 2 7B, but cannot transfer to Mistral 7B)
> - O2P did not report numbers for Llama 2 13B in their original work, so we only use their reported Mistral 7B results. We do not believe that adding this would be particularly informative or change our conclusion that O2P outperforms our method in the transfer setting.

---

> > ### Comment · Reviewer_uYfj · 2025-08-04
> >
> > Thanks for the response. It would be good if you could clarify the minor points we discussed in the final version of the paper.

---

### Official Review · Reviewer_QJEH · 2025-07-03

**Clarity:** 3
**Significance:** 3
**Originality:** 3
**Rating:** 5
**Confidence:** 4

**Summary:**

This work presents a language model inversion (i.e. reconstructing information on the inputs of the model based on its behavior) which relies on sequences of compressed logprobs (i.e. not requiring the full logprobs vectors). This allows not only a faster way (in terms of number of queries) to construct the data used for the inversion, but also much better performances compared to previous work.

**Questions:**

- Typo line 96: "an language model"
- I am slightly confused regarding what your intuition for your approach to work could be. You write, line 227: "his is likely because post-training discourages target models from revealing system message". But it doesn't seem natural to me that information about the prompt couldn't be extracted from logprobs of responses that do not try to repeat the system prompt. What do you believe is the mechanism at play behind the success of logprob-based inversion (particularly yours which leverage sequences of logprobs)?
- Could you try and make your Figure 3 clearer? What exactly is given to the model and how does that evolve with the sequence of prediction? Why are there so many tokens with such a high probability? How should they be read? What does it mean for a token to be recoverable? Why do the filled red grid cells matter?
- Why do you think model providers still allow for access to models' logprobs?

**Ethical Concerns:**

["NO or VERY MINOR ethics concerns only"]

**Final Justification:**

The authors addressed my concerns and I found the discussions with other reviewers interesting too. I maintain my score.

**Limitations:**

Limitations seem addressed to me.

**Paper Formatting Concerns:**

No paper formatting concern.

**Quality:**

3

**Strengths And Weaknesses:**

## Clarity
### Strengths
- Paper is well written and very easy to follow
- Illustrations are clear and helpful
### Weaknesses
- The grid plots such as in Figure 3 are difficult to understand and parse.

## Quality
### Strengths
- The paper shows strong results with massive improvements compared to previous approach on certain experiments.
- The PILS approach seems practical as a way to extract user information contained in a prompt.
### Weakness
- The choice of models used should be justified, especially given that Llama 2 can be considered an old model, now.

## Significance & Originality
### Strengths
- This work demonstrates a practical language model inversion attack and seem to be the first to leverage working from a (transformed) sequence of logprobs to gain more information.
- The lossless compression approach suggested is interesting and seem to indicate not only a low-dimensional space of interest in which attacks can be successful, but also potential for transferability (as illustrated) across models.
### Weaknesses
- This work builds upon previous work (L2T) but the improvements are substantial enough to be considered novel.

---

> ### Author Rebuttal · Authors · 2025-07-31
>
> Thank you for your positive review and valuable feedback. We will make sure to improve the paper accordingly. We will address your questions and concerns as best we can below
>
> (Grid plots and figure 3)
> In figure 3, the target model receives the prompt (xtick labels) and generates an output (ytick labels) one token at a time. At each generation step, we feed the outputs *so far* to the inverter, and force it to generate the hidden prompt, measuring the probability of each prompt token according to the inverter. These probabilities are a single row of the grid. The rows with the most high probability tokens (mostly gray) are the ones where the model is closest to correctly guessing the hidden prompt.  Token is “recoverable” when its probability is high. Filled red grid cells mean that the target model had to generate the hidden token before the inverter was able to guess what it was. We will try to make these figures easier to parse in the final draft.
>
> (Choice of model) We agree that Llama 2 is a weaker model, so we also provide Llama 3 results in Tables 1 and 2 for the benefit of future work. We used Llama 2 models in order to make our results comparable to previous work without replicating all of their experiments on Llama 3. We will make sure that this is explicit in the final draft.
>
> (Intuition for why our method works) You are correct in that logprob-based inversion can recover the prompt even when it does not appear in the text outputs of the model. Line 227 is only trying to convey that inversion success appears to be correlated with how likely the model is to generate the secret prompt, even for logprob based methods. As to the mechanism for how our method excels, compared to text-based inversion, logprob-based inversion accesses richer information about the prompt directly from the model’s internal representation. Compared to the previous logprob-based inversion, our method gives the inverter access to information that only surfaces later in the generation process. We are happy to discuss more about these intuitions.
>
> (Why do model providers allow logprob access) We are not sure about all the reasons, but we presume that logprobs and logit bias are useful to API users for one reason or another (e.g., fine-grained control of generations), and so removing these options from the API would cost providers in some way. It may be that these costs outweigh the risks of providing logprobs, at least for now.

---

> > ### Comment · Reviewer_QJEH · 2025-08-04
> >
> > Thank you for providing clarification in this rebuttal. I read it with great attention.
> >
> > Regarding Figure 3, your explanations are helping. It seems like the explanation for this plot could be improved in the manuscript. Allow me to suggest:
> > - Change the colormapping to one easier to read. Shades of gray are hard to differentiate and it's not clear what is the threshold (numerically and visually) for a token to be considered "recoverable" or not.
> > - Why is it important that the inverter can cannot recover a token before the target model outputs it?
> >
> > You addressed my concerns and I would appreciate if you could improve the figures similar to Figure 3 with better colormapping and clearer explanations in the manuscript.

---

> > > ### Author Response · Authors · 2025-08-04
> > >
> > > We appreciate your reply and engagement with our rebuttal. We will be sure to update the figures in the final draft.

---

### Official Review · Reviewer_sEEm · 2025-07-03

**Clarity:** 3
**Significance:** 4
**Originality:** 4
**Rating:** 5
**Confidence:** 4

**Summary:**

This paper introduces Prompt Inversion from Logprob Sequences (PILS), a method to recover hidden prompts from a language model. The key contribution is a novel technique to losslessly compress sequences of next-token probability distributions into compact, low-dimensional vectors using the Additive Log-Ratio (ALR) transform. This allows the inverter to leverage information from multiple generation steps, a significant departure from prior work. PILS achieves state-of-the-art results, improving exact prompt recovery rates by 2-3.5x over previous methods. The paper also reports a surprising "length generalization" capability, where the inverter's performance improves on longer sequences than it was trained on.

**Questions:**

1.  **ALR Implementation:** Your use of a random token set for the ALR transform seems ad-hoc. How sensitive is performance to this choice, and did you evaluate more standard, stable alternatives like Centered or Isometric Log-Ratio (CLR/ILR) transforms?[2, 3]
2.  **Model Transfer:** The proposed transfer method underperforms text-based baselines significantly. Why are logprob-based features less transferable? Have you considered more advanced alignment techniques like Centered Kernel Alignment (CKA)?[4]
3.  **Length Generalization:** To substantiate your claim that relative position embeddings (RPEs) cause length generalization, can you provide an ablation study where an inverter with absolute position embeddings fails to generalize?
4.  **Practical Cost:** Can you provide a concrete cost analysis (in dollars and time) for executing a PILS attack against a commercial API, factoring in the bisection search for logprobs?

**Ethical Concerns:**

["NO or VERY MINOR ethics concerns only"]

**Limitations:**

Yes

**Quality:**

4

**Strengths And Weaknesses:**

**Strengths:**
1.  **State-of-the-Art Performance:** The method achieves a 2-3.5x improvement in exact prompt recovery over strong baselines, a significant leap that elevates logprob-based inversion from a theoretical to a practical security threat.
2.  **Novelty and Technical Contribution:** The core ideas are highly original: inverting a *sequence* of probability distributions and using a principled, lossless compression (ALR transform) to make this feasible. The discovery of length generalization is also a valuable scientific finding.
3.  **Rigorous Evaluation:** The claims are supported by comprehensive experiments against multiple baselines, across several LLMs and diverse datasets, including insightful qualitative analysis.

**Weaknesses:**
1.  **Poor Model Transfer:** The proposed method for transferring the inverter to a new model family performs poorly, significantly lagging behind simpler text-based attacks. This limits the attack's scalability, as a new inverter must be trained for each target model family.
2.  **Ad-Hoc Compression Implementation:** The choice to use a random set of tokens as a basis for the ALR transform to avoid "degenerate compression" is not well-justified and lacks analysis of alternatives (e.g., CLR, ILR), raising questions about the method's robustness.[1, 2, 3]

---

> ### Author Rebuttal · Authors · 2025-07-31
>
> Thank you for your positive score and valuable feedback. We appreciate your comments and will use them to improve our paper. Regarding the listed weaknesses, we hope you agree that poor model transfer is a limitation of the method but not a weakness of the paper, since an honest evaluation of the limitations should be part of any analysis of a new method. See the answer below to question 2 for more details on our transfer experiments. Similarly, we believe that our answer below to your question 1 addresses the second listed weakness.
>
> Questions:
>
> 1. (ALR implementation) In our preliminary experiments, we observed that tokens whose corresponding unembedding submatrices are low-rank consistently underperformed compared to those with full-rank submatrices. This makes sense because linearly dependent tokens provide redundant information about the hidden state. Empirically, we find that the first D tokens tend to be low rank (this may depend on the model), but a random choice of D+100 tokens usually has high rank, and therefore performs well. We do not observe sensitivity to the choice of random tokens, and choosing random tokens is much easier than searching for a full rank submatrix. We also explored using other transforms like CLR, but observed no significant performance improvements over our proposed approach. We adopted ALR primarily due to its efficiency advantages: while CLR requires computing the mean over the entire logprob vector (necessitating access to all vocabulary tokens), ALR operates with only D+101 logprobs, which is substantially smaller. This design choice significantly reduces API costs (as measured in logprobs) without any impact on performance.
>
> 2. (Model Transfer) We experimented with several variations on our logprob alignment technique and did not find any improvements, though this does not rule out the possibility that other techniques like CKA might work. We report these findings despite their limitations because our method is the first ever cross-family logprob-based inverter transfer method, and could precipitate better results in future work. We speculate in the paper (line 257) that transfer suffers because the inverter “overfits” to model-specific features from the source model that are not present in logprobs from other models.
>
> 3. (Positional embedding generalization ablation) This would be an interesting experiment to run, though the rebuttal period is not long enough to accommodate it. We will consider adding this for the camera ready. Since this claim is speculation, and not central to the findings of the paper, we can consider removing it, though we think that it is an interesting avenue for future work.
>
> 4. (Practical Cost): Great idea, we will add this to the paper. Using some back-of-the-envelope calculations, let’s calculate the cost for a T-step attack on an API model with hidden size D, max logit bias B.
> We need $T\times D$ logprobs, each logprob requires $\log(B/\epsilon)$ queries. Each query has up to $T$ input tokens and 1 output token. Suppose the input cost is $C_\text{in}$ per token and the output token cost is $C_\text{out}$ per token. So the cost will be
> $$\sum_{i=0}^{T-1}D(i\times C_\text{in}+C_\text{out})\log(B/\epsilon).$$
> If we are using an GPT-4.1 mini (one size up from the cheapest model GPT-4.1 nano), this model has an unknown hidden size D, but perhaps we assume it is similar to GPT-3.5-Turbo which is less than 4600. Max logit bias is 100, cost is $0.40/1M tokens and cost per output is $1.60/1M tokens, and we will ignore caching, though that could reduce cost further. We will require logprobs with precision of 0.001.
> This gives us
> $$\sum_{i=0}^{15}4600\times(i\times\frac{0.1}{1000000}+\frac{0.4}{1000000})\log_2(100/0.001)=\$5.50$$.
> For GPT-4.1 nano, this cost is $1.38.
> Using the batch API, both of these costs can be halved.
> This cost per prompt is relatively low, and creating a small dataset to fine-tune an inverter model to target GPT-4.1 would be feasible.

---

### Comment · Area_Chair_nQXd · 2025-08-06
**Please read the authors’ rebuttal and join the discussion**

Dear Reviewers sEEm and vtuB,

Thank you for your valuable contributions.

The authors have provided rebuttal to your reviews. Please carefully read their responses as soon as possible, and indicate whether your concerns have been addressed. You are also encouraged to engage in discussion with fellow reviewers.

Best,

AC

---

### Decision · Program_Chairs · 2025-09-17

**Decision:**

Accept (poster)

**Comment:**

This paper studies language model inversion and proposes compact representations of next-token distributions to improve input reconstruction. The approach is simple and clearly motivated, with theoretical analysis and empirical validation across model scales and datasets. Reviewers appreciated the clarity of writing and the technical soundness, while raising concerns about the breadth of evaluation and real-world impact. The rebuttal addressed methodological questions and provided additional analysis, which satisfied most reviewers. While the contribution is incremental in scope, it is technically solid, easy-to-implement, and could be of interest for the community of inversion attacks. Overall, I recommend acceptance for this paper.